# Standardisation and Optimisation of Chest and Pelvis X-Ray Imaging Protocols Across Multiple Radiography Systems in a Radiology Department

**DOI:** 10.3390/diagnostics15121450

**Published:** 2025-06-06

**Authors:** Ahmed Jibril Abdi, Kasper Rørdam Jensen, Pia Iben Pietersen, Janni Jensen, Rune Lau Hovgaard, Ask Kristian Aas Holmboe, Sofie Gregersen

**Affiliations:** 1Department of Clinical Engineering, Region of Southern Denmark, Odense University Hospital, J. B. Winsløws Vej 4, 5000 Odense, Denmark; 2Research and Innovation Unit of Radiology, University of Southern Denmark, Kløvervænget 10, 5000 Odense, Denmark; 3Department of Radiology, Odense University Hospital, J. B. Winsløws Vej 4, 5000 Odense, Denmark; 4CAI-X (Centre for Clinical Artificial Intelligence), Odense University Hospital, University of Southern Denmark, 5000 Odense, Denmark; 5Department of Radiology, Odense University Hospital, Baagøes Allé 15, 5700 Svendborg, Denmark

**Keywords:** protocol optimisation, chest radiography, pelvis radiography, visual grading analysis (VGA), quantitative image quality

## Abstract

X-ray imaging protocols in radiology departments often exhibit variability in exposure parameters and geometric setups, leading to inconsistencies in image quality and potential variations in patient dose. **Objectives**: This study aimed to harmonise and optimise chest and pelvis X-ray imaging protocols by standardising exposure parameters and geometric setups across departmental systems, minimising radiation dose while ensuring adequate image quality for accurate diagnosis. **Methods**: The image quality of five pelvic and three chest protocols across different radiographic systems was evaluated both quantitatively and visually. Visual image quality for both chest and pelvis protocols was assessed by radiologists and radiographers using the Visual Grading Analysis (VGA) method. Additionally, the quantitative image quality figure inverse (*IQF_inv_*) metric for all protocols was determined using the CDRAD image quality phantom. Moreover, the patient radiation dose for both chest and pelvis protocols was evaluated using dose area product (DAP) values measured by the systems’ built-in DAP metres. **Results**: Different quantitative image quality and radiation dose to patients were achieved in various protocol settings for both chest and pelvis examinations, but the visual image quality assessment showed satisfactory image quality for all observers in both the pelvis and chest protocols. The selected protocols for harmonising chest radiography across all imaging systems result in reduced radiation exposure for patients while maintaining adequate image quality compared to the previously used system-specific protocol. **Conclusions**: The clinical protocol for chest and pelvis radiography has been standardised and optimised in accordance with patient radiation exposure and image quality. This approach aligns with the ALARA (As Low As Reasonably Achievable) principle, ensuring optimal diagnostic information while minimising the radiation risks.

## 1. Introduction

X-ray imaging utilises ionising radiation, which can interact with human cells and potentially induce biological damage [1,2]. While acknowledging this risk, X-ray imaging remains indispensable for the diagnosis of numerous medical conditions and plays a critical role in patient care and life-saving interventions [3,4]. A comprehensive risk–benefit assessment consistently demonstrates that the diagnostic benefits of X-ray imaging outweigh the associated radiation risks [5,6,7].

Within the field of radiation protection, the optimisation principle of “As Low As Reasonably Achievable” (ALARA), which prioritises achieving adequate image quality while simultaneously minimising radiation exposure, has served as a fundamental guideline for decades [8]. This principle, initially promulgated by the International Commission on Radiological Protection (ICRP), has been widely adopted as a regulatory standard in many countries worldwide [9].

The diagnostic information and anatomical details obtained from a radiological examination are heavily influenced by the properties of the imaging system, the configuration of the exposure parameters, and the radiation dose applied [10]. The use of anthropomorphic phantoms facilitates the optimisation of clinical protocols without the need for patient-specific adjustments [11]. This approach allows for the minimisation of radiation exposure to patients while maintaining adequate image quality [12]. To optimise protocols, numerous parameters related to the radiographic system, particularly within the imaging chain, must be taken into account. These include tube voltage, dose selection, and tube filtration. Other parameters that play a crucial role in optimising the protocol are the geometric settings, including the source-to-image distance (SID) and the associated anti-scatter mechanisms. The type of detector/receptor and image processing packages are also important factors that can significantly influence the optimisation and standardisation of clinical protocols [13]. The pelvis is one of the radiographic examinations associated with high radiation doses to the patient, as several dose-sensitive organs, including the gonads and ovaries, are involved [14]. Therefore, it is important to use an optimal dose that provides sufficient diagnostic information in accordance with the ALARA principle. Chest radiographic examinations are common and widely used to detect lung diseases and anatomical changes in chest organs, with their use increasing. Thus, they provide extensive health information [15,16].

In a radiology department at Odense University Hospital, Svendborg, Denmark, equipped with six digital radiography (DR) systems from the same vendor used for both chest and pelvis examinations, a need for protocol optimisation was identified. Here, a substantial variation in protocol settings for both chest and pelvis radiography was observed by the clinical staff, arising from a range of clinical and technical rationales. The variation encompassed exposure parameters and geometric configuration. Certain chest protocols employed an extended source-to-image distance (SID) of 300 cm, a geometrical setup originating from historical film-screen radiography. Other protocol settings were based on default configurations provided by the equipment manufacturer, while some reflected legacy practices that had persisted within the department over time.

Variations in protocol configurations can result in differences in patient radiation dose as well as image quality.

The aim of this study is to standardise clinical protocols for chest and pelvic radiography to optimise diagnostic image quality while maintaining patient radiation doses within acceptable limits. This was achieved through the application of both objective image quality metrics and subjective visual assessment methods. Although the protocol standardisation was undertaken within a single radiology centre, the methodology and resulting protocols have the potential to inform practices in other radiology departments with comparable imaging systems and clinical practices.

## 2. Materials and Methods

Clinical protocols for chest and pelvis examinations were harmonised and optimised across six radiography systems. As part of this process, the optimisation and standardisation of the chest and pelvis protocols were undertaken. The investigation assessed potential technological, hardware, and software differences between these Siemens systems (Siemens Healtineers, Forchheim, Germany). All systems were found to operate on the same software platform, FluoroSpot Compact (FSC) (Siemens Healtineers, Forchheim, Germany), and utilise the same image processing software, DaimonView (Siemens Healtineers, Forchheim, Germany).

To minimise technical variability, radiographic systems were selected based on shared core components, including identical X-ray tubes, detector technology, operating platforms, and image processing software, which are known to affect image quality and radiation dose. Despite this standardisation, variations in exposure parameters and geometric configurations were found in the different protocols. Therefore, the protocols were standardised and optimised to ensure consistent diagnostic image quality and dose efficiency across all systems.

The acquisition parameter settings and technical specifications of all Siemens radiographic systems, which may influence patient radiation exposure and image quality, were retrieved from the electronic data sheet available on the imaging system console. These details are summarised in Table 1.

### 2.1. Protocol Settings and Image Acquisition

Ten technical phantom and ten clinical phantom images were acquired for each of the three technical and three clinical chest protocol settings, as well as ten images for each of the five technical and ten images for each of the three clinical pelvic protocol settings. This resulted in a total of 80 technical phantom images and 60 clinical phantom images across both protocols.

The protocols listed in Table 2 are the original protocol settings for both chest and pelvis, except for chest protocol setting 2. Chest protocol setting 2 is an optimised protocol developed during a previous thesis project at the Department of Radiology at Odense University Hospital with the aim of optimising and standardising chest imaging in terms of image quality and patient radiation dose [17], while chest protocol 1 and 3 represent the department’s original protocols.

The image quality of both chest and pelvis protocols was evaluated using quantitative and visual assessment methods. The quantitative image quality and radiation dose for all three chest and five pelvis protocols were assessed.

For all chest and pelvis protocol settings, the tube current was automatically adjusted by the automatic exposure control (AEC) system based on patient size. This approach reflects clinical routine and supports dose optimisation within the study.

### 2.2. Quantitative Image Quality Assessment

To evaluate the quantitative image quality for both chest and pelvis clinical protocols, the CDRAD 2.0 image quality phantom for conventional DR systems, supplemented with 20 cm PMMA plates to simulate the average patient size, was used [18].

PMMA plates used as a phantom in diagnostic radiography typically simulate an average adult with a thickness of 20–23 cm, corresponding to a body weight of approximately 70–75 kg [19,20]. This thickness approximates the anterior–posterior (AP) dimension of an adult torso and is also commonly used for quality assurance, dosimetry, and equipment calibration.

The CDRAD phantom consists of a 15 × 15 array of cells with cylindrical holes of varying sizes (0.3 to 8.0 mm) and depths. The depth is constant within each column, and the area is constant within each row. Figure 1 provides a photographic representation and an X-ray image of the CDRAD 2.0 phantom.

The CDRAD phantom provides a quantitative assessment of the system’s image quality, and the dose area product (DAP) was recorded for each protocol setup to evaluate the radiation dose. The CDRAD image quality was analysed and quantified using CDRAD analysis software. The CDRAD analyser software 2.0 derives two image quality parameters from phantom images: the Image Quality Figure Inverse (*IQF_inv_*), which quantifies overall image contrast, and a graphical representation of contrast–detail resolution. The *IQF_inv_* is defined as the summation of the product of contrast depth (*C_i_*) and the corresponding detectable diameter (*D_i_*_,*th*_) across all 15 columns of the phantom [18,21,22].$${IQF}_{inv}=\frac{100}{\sum_{i=1}^{15}C_{i}D_{i,th}}$$

The line connecting the central spots with the smallest visible diameter and contrast is known as the contrast–detail (CD) curve. These curves are generated by the CDRAD analysis software, with the x-axis representing the depths of the detected circles and the y-axis representing their diameters. This method evaluates the spatial and contrast resolution of an imaging system by determining the CD curves. Patient radiation dose was assessed using the system’s DAP, which undergoes quality control as part of the annual statutory test and monthly consistency tests.

### 2.3. Visual Image Quality Assessment

A LungMan phantom (Kyoto Kagaku, Kyoto, Japan), a versatile anthropomorphic chest model with dimensions of 43 × 40 × 48 cm, a weight of 18 kg, and a chest circumference of 94 cm, was used to assess the visual image quality for the chest protocol settings [23]. The phantom is designed for use in both conventional radiography and CT imaging for protocol optimisation and includes two removable fat-equivalent layers, each 27 mm thick, allowing the simulation of an additional total body thickness of 54 mm [13].

A pelvis phantom representing an average adult male (175 cm tall, 74 kg), and containing human bone, was used to evaluate visual image quality for the pelvic protocol settings.

The phantoms are depicted in the following figures, with the chest LungMan phantom shown in Figure 2 and the pelvic phantom shown in Figure 3.

The local medical physicists, the super-user radiographers of the imaging systems, and the department heads discussed and evaluated the preliminary results of the quantitative image quality. Through consensus, pelvis protocol settings 2 and 4 were excluded from the visual image quality assessment segment. These two excluded pelvis protocol settings have the highest device dose/speed setting of 5 µGy, which has resulted in the highest radiation doses to patients. However, the radiation doses for these two protocol settings remain below the national diagnostic reference levels (DRLs). These protocol settings were excluded not only due to their relatively higher radiation exposure but also to reduce the total number of images requiring visual assessment. Assessing the clinical images of all five pelvis protocol settings would have been considerably more time-consuming for the radiographers responsible for image evaluation.

To achieve a more optimised and consistent approach, a visual assessment of image quality was conducted for the three selected protocol settings in both the chest and pelvic protocols, as shown in Table 3.

Chest protocol 3, which uses a higher SID of 300 cm, originates from the film-screen era and is not ideally suited to digital detectors [24], although this geometrical setting is still available on some systems in the department.

A senior radiologist and a departmental specialist radiologist, both with a minimum of 5 years of experience, visually evaluated the chest protocols, while two experienced reporting radiographers evaluated the visual image quality of the pelvic protocols.

Visual image quality assessment for both the chest and pelvic protocols was conducted on a Picture Archiving and Communication System (PACS) workstation with a diagnostic monitor under relevant lighting and environmental conditions, consistent with routine clinical practice.

The diagnostic monitors used for scoring images from both the chest and pelvis examinations met the diagnostic resolution requirements, with a resolution of 5 megapixels. The monitors were subject to mandatory quality assurance procedures, including monthly status checks performed using an internal sensor, with quarterly visual consistency tests and regular calibration carried out as needed.

The Viewer for Digital Evaluation of X-ray Images (ViewDEX) Software v2.48 was used to assess and score the image quality across all imaging systems and clinical examination configurations [25,26]. ViewDEX is a Java-based application designed for displaying and assessing medical images in observer performance studies, as well as for analysing and scoring the quality of radiological images [27].

A Visual Grading Analysis (VGA) was conducted to evaluate the visual image quality of the protocols. A VGA is a statistical approach commonly used in fields such as radiology to assess image quality by comparing images to a reference standard through observer scores [28]. VGA is divided into two methods for analysing image quality: absolute VGA and relative VGA [29].

In this study, the absolute VGA method was used to evaluate visual image quality for both the chest and pelvic protocols. In this approach, each image was evaluated independently of the observers using predefined image quality criteria such as clarity and anatomical detail without reference to a standard image or a direct comparison with other images. To minimise potential bias, all images were randomised, and observers were blinded to the protocol settings for each image.

It quantifies subjective evaluations, such as clarity and detail visibility, by calculating the mean score and variance across observers. A standard equation for *VGA* can be formulated as follows [21,28].$$VGAscore=\frac{1}{n}\sum_{i=1}^{n}S_{i}$$
where *S_i_* represents the score given by observer *i*, and *n* is the total number of observers.

The predefined rating scale shown in Table 4 was used for the visual evaluation of image quality for both chest and pelvis protocols. Scores range from 1 (poor image quality) to 5 (excellent image quality) [28,29,30].

The image quality criteria for chest and pelvis examinations are based on European guidelines but were also adapted by the observers to their routine diagnostic interpretation in daily practice [31,32].

The VGA image quality criteria for chest protocols are shown in Table 5. The corresponding VAG image quality criteria for the pelvis protocol are shown in Table 6.

### 2.4. Statistical Analysis Method

The Kruskal–Wallis test, a non-parametric method, was used to assess the differences between the groups based on the protocol settings. A significance level of *p* < 0.05 was applied. Additionally, 95% confidence intervals for the median differences were calculated to quantify the uncertainty in the estimates.

This test statistic was chosen since the data for the evaluation of quantitative image quality were not normally distributed.

Statistical analyses were conducted using the Statistical Package for the Social Sciences (SPSS), Release 26.0.0.0 (IBM Corp., Armonk, NY, USA).

## 3. Results

### 3.1. Quantitative Image Quality and Radiation Dose Results

The quantitative image quality metric, *IQF_inv_*, and the patient’s radiation dose, indicated by the DAP, were evaluated and are presented in Table 7 for all the initial pelvic and chest protocol settings.

Comparing the radiation doses and *IQF_inv_* of the chest protocols, we find that protocol setting 1 has the best *IQF_inv_* score but at the cost of a high radiation dose compared to protocols 2 and 3. The radiation doses of protocols 2 and 3 are similar, but protocol setting 2 has a superior *IQF_inv_*, only slightly less than for protocol 1.

For the pelvis protocols, protocol settings 2 and 4 yield the highest radiation doses and the best quantitative image quality metric *IQF_inv_*. Protocol settings 1 and 5 have nearly identical radiation doses, but protocol setting 5 has a superior quantitative image quality metric *IQF_inv_*. Protocol setting 3 results in the lowest radiation dose while maintaining a reasonable quantitative image quality metric *IQF_inv_*.

The contrast–detail curves (CDCs) for three chest radiography protocol settings and five pelvis radiography protocol settings are shown in Figure 4. CDCs describe the relationship between the minimum object size and the minimum contrast required for the detection of objects in medical imaging. These curves offer insights into the minimum contrast necessary to visualise objects of different sizes.

Contrast–detail curves positioned closer to the lower left corner of the coordinate system indicate improved image quality, as they correspond to the detection of smaller objects at lower contrast levels [33]. Thus, protocols with contrast–detail curves near the lower left corner of the axes represent superior image quality.

### 3.2. Visual Image Assessment Results

The visual assessment of the image quality for the three selected chest protocol settings, with VGA score results, is shown in Table 8. The average VGA score for the chest protocols given by both radiologists is quite high (close to the maximum score), ranging from 4.72 to 4.96. This indicates that the visual image quality is excellent for all five criteria and that the diagnostic use of the images is not limited by any of the protocol settings. The corresponding VGA results for the pelvic protocol settings are presented in Table 9.

The results presented in Table 9 indicate that both reporting radiographers assigned the highest possible VGA score of 5 to the pelvis protocols in all image quality criteria.

### 3.3. Statistical Analysis Results

A Kruskal–Wallis statistical comparison was conducted for the quantitative evaluation of image quality and patient radiation dose across all protocol settings. However, given that all the observers consistently assigned the highest possible scores across all protocol settings and associated criteria for both chest and pelvis examinations, demonstrating minimal variation, a statistical analysis of inter- and intra-observer variability was not conducted.

#### 3.3.1. Statistical Comparison of the Radiation Doses

The Kruskal–Wallis statistical comparison of the DAP values across the chest and pelvis protocol settings is shown in Table 10. The “Test Statistic” column reflects the difference between the mean values of each protocol pair. The “Standard Error” and “Standardised Test Statistic” columns provide insights into the statistical significance of these mean differences. The “*p*-values” column indicates the probability that the observed differences occurred by chance.

For the DAP values in the chest protocol settings, significant differences (*p* < 0.05) were found in protocol setting 3 vs. 2, protocol setting 3 vs. 1, and protocol setting 2 vs. 1, with the largest difference in protocol setting 3 vs. 1 (*p* < 0.001).

For the DAP values in the pelvis protocol settings, most comparisons showed significant differences (*p* < 0.05), except protocol setting 3 vs. 5, protocol setting 5 vs. 1, protocol setting 1 vs. 4, and protocol setting 4 vs. 2 (*p* > 0.05). The largest differences were in protocol setting 3 vs. 2, protocol setting 3 vs. 4, and protocol setting 5 vs. 2 (*p* < 0.001). Protocol setting 3 had the strongest distinctions, while protocol settings 5 and 4 had fewer significant differences.

#### 3.3.2. Statistical Comparison of the Quantitative Image Quality Metric, *IQF_inv_*

The Kruskal–Wallis statistical comparison of the quantitative image quality metric IQF_inv_ was conducted to assess variations across protocol settings, with the pairwise comparisons of chest protocol settings presented in Table 11.

Statistical analysis revealed no significant difference in the *IQF_inv_* between protocol settings 1 and 2 (*p* = 0.220). In contrast, significant differences were observed between protocol settings 1 and 3 (*p* < 0.001) and between protocol settings 2 and 3 (*p* = 0.002). These findings suggest that protocol settings 1 and 2 delivered comparable image quality, whereas protocol setting 3 resulted in notably lower image quality.

A pairwise boxplot comparing three different chest protocol settings is illustrated in Figure 5. The boxplot shows that protocol setting 1 has the highest *IQF_inv_*, indicating the best image quality, followed by protocol setting 2, while protocol setting 3 has the lowest values with greater variability and an outlier. This suggests that protocol setting 1 provides the most consistent and superior image quality, whereas protocol setting 3 may compromise image quality, potentially due to dose reduction or different acquisition settings.

The results of the pairwise comparisons of the *IQF_inv_* across different pelvic protocol settings are presented in Table 12. As shown in Table 12, several statistically significant differences were identified between protocols, as indicated by *p*-values < 0.05. For instance, significant differences were found in most protocol comparisons (*p* < 0.05), except for protocol setting 1 vs. 3, protocol setting 5 vs. 2, protocol setting 5 vs. 4, and protocol setting 2 vs. 4 (*p* > 0.05). The largest differences were in protocol setting 1 vs. 2, protocol setting 1 vs. 4, and protocol setting 3 vs. 4 (*p* < 0.001). Protocol settings 1, 2, 3, and 4 show strong distinctions, while protocol setting 5 does not significantly differ from protocol settings 2 and 4.

A pairwise comparison of the *IQF_inv_* for the different pelvic protocol settings is shown in the boxplot in Figure 6. This plot illustrates the results of an independent-sample Kruskal–Wallis test on the image quality measured by the *IQF_inv_*. The Kruskal–Wallis test shows statistically significant differences between the medians of these protocols. Since protocol 2 has consistently higher *IQF_inv_* values, it can be concluded that there are significant differences in the image quality between protocol 2 and the other protocol settings.

## 4. Discussion

This study undertook a comprehensive standardisation of five pelvic protocols, each with varying exposure parameters and geometric setups, as well as three chest protocols, with differences in both geometry and exposure parameters. Previously used, non-optimised protocol settings for both chest and pelvis imaging revealed notable variations in patient dose and image quality, raising concerns among radiologists about inconsistencies in image quality across imaging systems. These findings highlighted the need for harmonisation to ensure both diagnostic reliability and optimised patient safety.

Protocol setting 2 was selected for harmonising chest protocols due to its practicality and alignment with current standards [32], despite not achieving the lowest possible radiation dose.

While protocol setting 3, which resulted in the lowest radiation dose, demonstrated sufficient visual and quantitative image quality, it was not adopted due to its geometric configuration. Specifically, the SID in protocol setting 3 does not conform to the current standards outlined in the European Guidelines [32,34], as it is based on film-screen imaging systems [35]. Additionally, chest protocol setting 3 poses logistical challenges, as its geometric requirements necessitate a larger room area, complicating installation and the design of chest radiography facilities. This limitation is particularly relevant in modern clinical settings, where efficient space utilisation is essential.

The selection of the chest protocol setting 2, with the second-lowest radiation dose, represents a balanced approach that prioritises patient safety through dose reduction while also considering the practicality of implementation within the constraints of modern equipment and facility capabilities. This decision underscores the necessity of a multifaceted approach to protocol standardisation, incorporating not only technical performance but also logistical and practical considerations in the decision-making process.

Pelvis protocol setting 3 was selected as the standardised protocol within the department based on its combination of the lowest patient radiation dose, a comparatively higher quantitative image quality metric, *IQF_inv_*, and the highest VGA scoring assigned by both observers. The fact that all pelvis protocols received perfect visual grading scores indicates that, from a subjective visual assessment standpoint, image quality was perceived as equivalent across the protocols. Therefore, visual quality alone could not serve as a distinguishing factor. This reinforced the importance of incorporating objective dose and quantitative image quality metrics in protocol selection, as protocol setting 3 provided the most favourable balance between the diagnostic quality and radiation protection.

The consistently high VGA scores observed for the pelvis protocol settings may reflect a ceiling effect in which the scoring scale is not sensitive enough to detect incremental differences in image quality. This limitation could reduce the ability to discriminate between protocol settings. In addition, the low inter-observer variation may have contributed to the uniformity of the results. These factors suggest that future studies should include a wider range of protocol variations or include a larger number of observers to increase methodological sensitivity and improve the discriminative power of image quality assessment.

In pelvic radiography, its higher spatial resolution enhances the visualisation of anatomical structures, supporting accurate image interpretation and clinical assessment [36].

Adopting a harmonised protocol enhances consistency across departmental systems, reducing variability in imaging outcomes. This consistency is important for comparative studies, interdepartmental collaboration, and adherence to regulatory standards. While pelvis protocol setting 3 demonstrates excellent performance in terms of dose and image quality, ongoing evaluation is essential. Regular audits and feedback from radiologists and technicians will ensure that the protocol continues to meet clinical demands and aligns with evolving guidelines and technological advancements.

The selected protocols for both chest and pelvis imaging incorporate a tube filtration of 0.2 mm of copper, which effectively attenuates low-energy X-rays that do not contribute to image formation but increase patient radiation exposure [37]. By filtering these softer X-rays, the protocols help minimise unnecessary radiation and reduce skin dose, thereby optimising radiation protection while maintaining diagnostic image quality [38].

The automatic exposure control (AEC) system is anticipated to perform consistently with the newly standardised protocols, as these protocols are designed within the operational range of the AEC detectors. Standardisation ensures more predictable exposure conditions, allowing the AEC to optimise the image receptor dose effectively across varying patient sizes. However, continuous monitoring is essential to ensure that AEC performance remains stable, particularly for outlier patient populations. Similar findings have been reported in the literature [39], emphasising the importance of aligning imaging protocols with AEC functionality to ensure dose consistency and maintain image quality.

The DAP values achieved in this study are considerably lower than the current national diagnostic reference levels (DRLs) for Denmark, for both chest and pelvis radiography. Specifically, the DAP values for the chest protocols ranged from 0.043 to 0.075 Gy·cm^2^, while those for pelvis protocols ranged from 0.18 to 0.43 Gy·cm^2^. In comparison, the Danish national DRLs for chest radiography are 0.3 Gy·cm^2^ and 1.5 Gy·cm^2^ for pelvis radiography [40]. These findings clearly indicate that the protocols employed in this study are well optimised in terms of radiation dose, offering substantial reductions without compromising diagnostic image quality. Such dose optimisation aligns with the principles of radiation protection and supports ongoing efforts to reduce patient exposure wherever feasible. The DAP values obtained in this study for both pelvis and chest radiography were lower than those reported in previous studies [41,42], indicating effective dose protocol optimisation. These findings support continued efforts to reduce the radiation dose in line with the ALARA principle.

The visual image quality assessment of the protocols was conducted using clinical anthropomorphic phantoms for both chest and pelvis protocols, which entail both advantages and limitations for protocol standardisation. The key limitation was that the selected phantom size might not fully represent the range of patient body sizes, despite both anthropomorphic phantoms approximating an average patient size. However, the integrated AEC systems in these protocols are expected to adjust the acquisition parameters based on patient size variations. Moreover, the use of these phantoms limits direct generalisation to the patient population.

Another potential limitation of this study was the apparent ceiling effect observed in the pelvis protocol assessments, where all observers consistently assigned the highest VGA scores. This may suggest limited sensitivity in distinguishing small variations in image quality within this protocol or a genuine consensus among observers due to high image quality. Additionally, the low variation between observers could reflect the use of clear, standardised scoring criteria, although it may also limit the granularity of the assessment.

A key advantage of this study was the stability and uniformity of clinical image quality assessment using phantom images, as they were free from motion artefacts caused by patients or their internal organs. Additionally, during the measurements, multiple phantom image acquisitions were conducted without concerns regarding patient radiation exposure.

This study establishes a foundation for the development of standardised clinical examination protocols that are applicable across radiography systems from different vendors and adaptable to various radiological departments. Future research will aim to expand the evaluation to include a wider range of anatomical regions and imaging modalities, as well as to incorporate clinical outcome measures to enhance the relevance and applicability of the findings. These efforts will support the establishment of consistent, patient-centred imaging practices within both scientific and clinical communities.

## 5. Conclusions

In collaboration with radiographers, radiologists, department management, and medical physicists, it was possible to standardise and optimise the pelvic and chest protocols for five digital radiographic systems. The harmonised protocols achieve the lower radiation dose to the patients while maintaining acceptable image quality for both technical and clinical assessment. The chosen protocol settings align with the ALARA principle, ensuring optimal balance between dose reduction and image quality.

The evaluation and validation of the selected and optimised protocol settings have been conducted in real patient populations to assess their applicability and strengthen their clinical relevance.

## Figures and Tables

**Figure 1 diagnostics-15-01450-f001:**
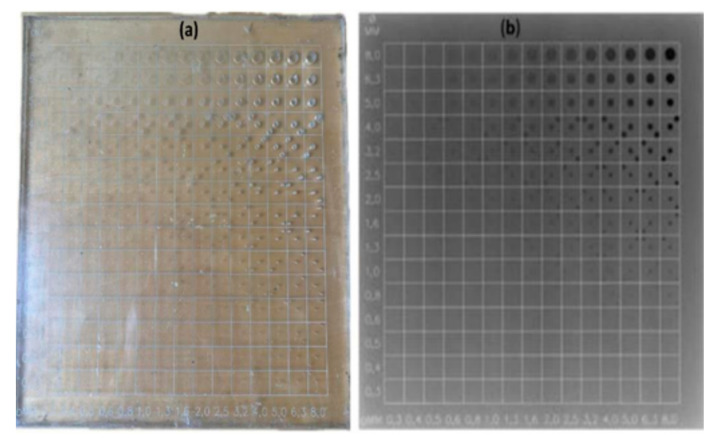
Photographic image of CDRAD phantom (**a**) and a radiographic image of the CDRAD phantom (**b**).

**Figure 2 diagnostics-15-01450-f002:**
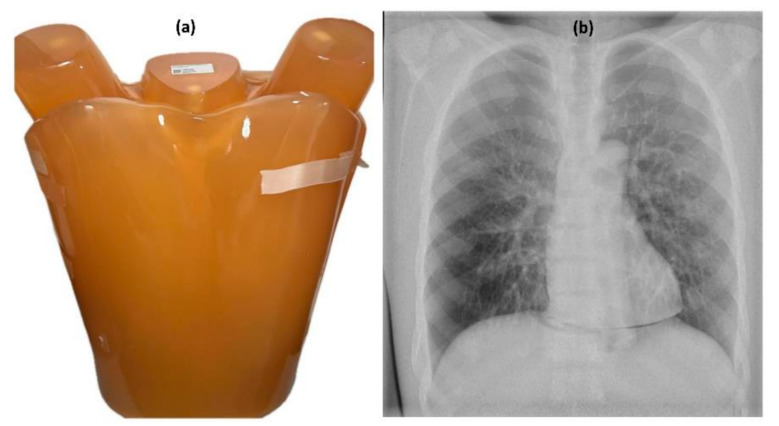
Photographic image of the LungMan anthropomorphic phantom (**a**) and X-ray image of the phantom (**b**).

**Figure 3 diagnostics-15-01450-f003:**
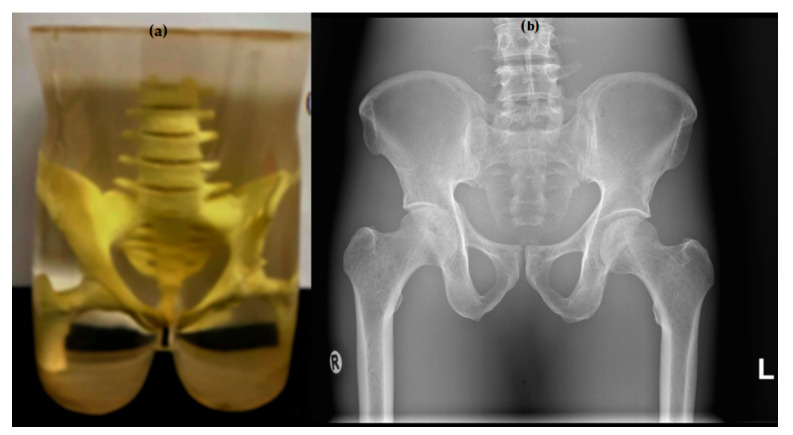
Photographic image of the pelvic phantom (**a**) and X-ray image of the pelvic phantom (**b**).

**Figure 4 diagnostics-15-01450-f004:**
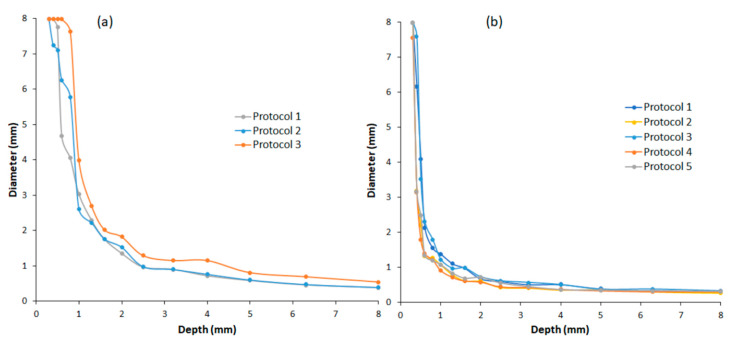
Contrast–detail curves for both chest (**a**) and pelvis (**b**) protocols for technical image quality evaluation.

**Figure 5 diagnostics-15-01450-f005:**
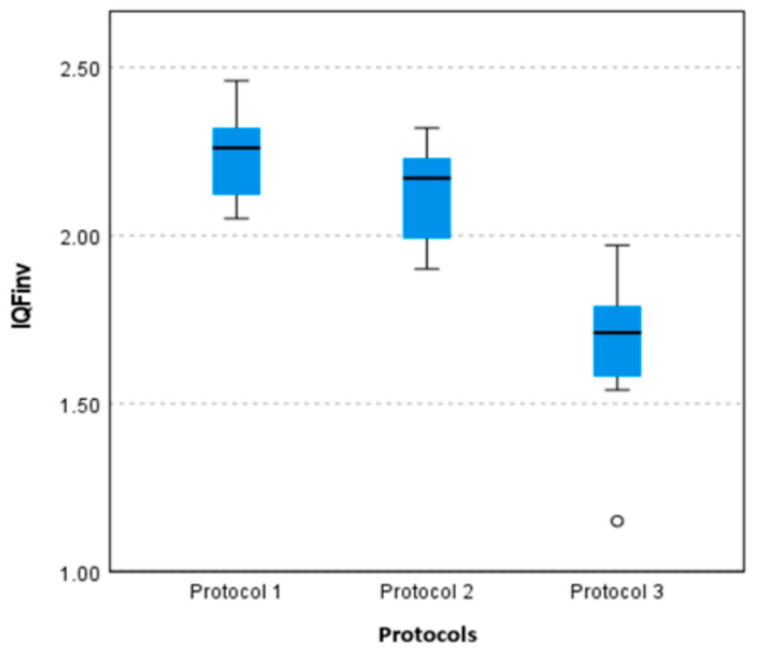
Boxplot comparison of the *IQF_inv_* for three chest protocol settings.

**Figure 6 diagnostics-15-01450-f006:**
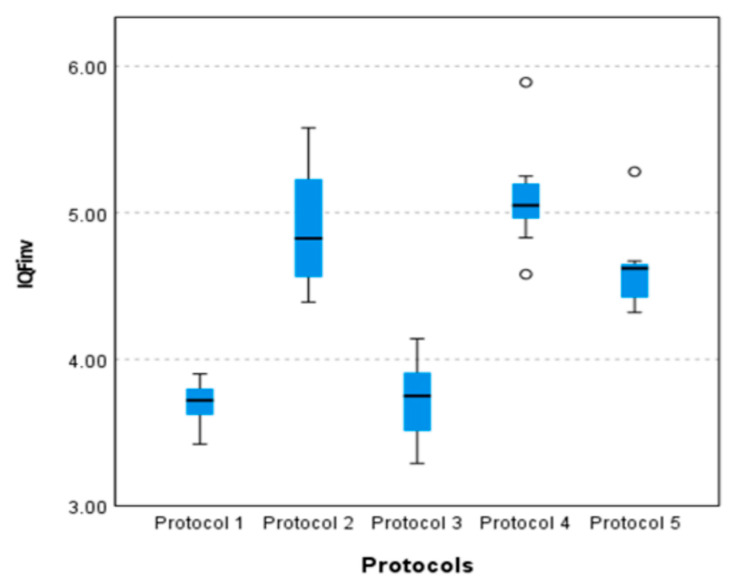
Boxplot comparison of the *IQF_inv_* for five pelvis protocol settings.

**Table 1 diagnostics-15-01450-t001:** Technical parameter settings and specifications of the radiographic systems.

Parameters	Specifications
Tube voltage (kV)	40–150
Tube current max (mA)	1000
Tube load (mAs)	0.5–800
Exposure time (s)	0.001–5
Generator power (kW)	80
Focal spot (mm)	0.6/1.0
Target angle (degree)	12
Detector type	CsI, a-Si TFT
Detector size (inch)	14 × 17
Detector pitch (μm)	148
Active matrix (pixel)	2354 × 2872
DQE at 1 lp/mm	50%
MTF at 1 lp/mm	61%

**Table 2 diagnostics-15-01450-t002:** Original protocol parameters and geometrical settings for both pelvis and chest protocols; SID = source image distance and Cu = copper.

# Protocol	Tube Voltage (kV_p_)	Tube Load (mAs)	Tube Current (mA)	Filtration (Cu)	Dose/Speed	SID (cm)	Protocols
Protocol 1	125	2	621	0.0	2.50	180	Chest
Protocol 2	125	2	686	0.2	2.50	180	
Protocol 3	150	2	594	0.3	3.57	300	
Protocol 1	81	5	831	0.0	2.50	115	Pelvis
Protocol 2	81	12	897	0.1	5.00	115	
Protocol 3	81	8	926	0.2	2.50	115	
Protocol 4	81	15	844	0.2	5.00	115	
Protocol 5	75	20	805	0.3	3.57	115	

**Table 3 diagnostics-15-01450-t003:** Protocol settings where visual image quality was assessed for both chest and pelvic protocols.

# Protocol	Tube Voltage (kVp)	Filtration (Cu)	Dose/Speed	SID (cm)	Protocol
Protocol 1	125	0.00	2.50	180	chest
Protocol 2	125	0.20	2.50	180	
Protocol 3	150	0.30	3.57	300	
Protocol 1	81	0.00	2.50	115	pelvis
Protocol 3	81	0.20	2.50	115	
Protocol 5	75	0.30	3.57	115	

**Table 4 diagnostics-15-01450-t004:** Visual image quality scoring scales for both chest and pelvis protocols.

Scale	Description	Interpretation
1	Poor image quality	The image is not usable and causes loss of information
2	Limited image quality	Limited relevance for clinical use and significant loss of information
3	Adequate image quality	Moderate restrictions for clinical use without significant loss of info
4	Good image quality	Minimal limitations for clinical use
5	Excellent image quality	No restrictions on clinical use

**Table 5 diagnostics-15-01450-t005:** VGA image quality criteria for the chest protocol.

# Criteria	Criteria (Visualisation of)
1	Pulmonary vasculature and peripheral vessels (PA—Posterior–Anterior, an X-ray taken from the back to the front of the body).
2	Trachea and proximal bronchi.
3	Borders of the heart and the aorta.
4	Diaphragm and lateral costophrenic angles (the areas where the rib cage meets the diaphragm).
5	The spine is visible through the shadow of the heart.

**Table 6 diagnostics-15-01450-t006:** VGA image quality criteria for the pelvis protocols.

# Criteria	Criteria
1	Symmetrical reproduction of the pelvis.
2	Visualisation of the sacrum and its intervertebral spaces.
3	Visualisation of sacroiliac joints.
4	Reproduction of the femoral neck, which should not be distorted by shortening or rotation.
5	Reproduction of spongiosa and corticalis as well as visualisation of the trochanters.

**Table 7 diagnostics-15-01450-t007:** Results of quantitative image quality and radiation dose to the patients for both pelvis and chest protocols with protocol parameter settings.

# Protocol	Tube Voltage (kV)	Tube Current (mA)	Filtration (Cu)	Dose/Speed	SID (cm)	*IQF_inv_*	DAP (dGycm^2^)	Protocols
Protocol 1	125	621	0.00	2.50	180	2.26	0.75	Chest
Protocol 2	125	686	0.20	2.50	180	2.17	0.46	
Protocol 3	150	594	0.30	3.75	300	1.69	0.43	
Protocol 1	81	831	0.00	2.50	115	3.72	2.98	Pelvis
Protocol 2	81	897	0.10	5.00	115	4.87	4.28	
Protocol 3	81	926	0.20	2.50	115	3.63	1.78	
Protocol 4	81	844	0.20	5.00	115	4.97	3.63	
Protocol 5	75	805	0.30	3.75	115	4.63	2.89	

**Table 8 diagnostics-15-01450-t008:** Visual image quality scoring (VGA) by two radiologists on the three chest protocol settings, scored by radiologist 1 (R1) and radiologist 2 (R2).

# Protocol		Criteria/VGA				Mean VGA	Raters
	1	2	3	4	5		
Protocol 1	4.70	4.90	5.00	5.00	4.50	4.82	R1
Protocol 2	4.90	4.90	5.00	5.00	4.80	4.92	
Protocol 3	4.90	4.90	4.80	5.00	4.60	4.84	
Protocol 1	5.00	4.90	5.00	5.00	4.90	4.96	R2
Protocol 2	4.90	4.80	4.70	5.00	4.60	4.80	
Protocol 3	4.80	4.50	4.90	5.00	4.40	4.72	

**Table 9 diagnostics-15-01450-t009:** Visual image quality scoring (VGA) by two reporting radiographers on the pelvic protocol, scored by radiographer 1 (R1) and radiographer 2 (R2).

# Protocols	Criteria/VGA					Mean VGA	Raters
	1	2	3	4	5		
Protocol 1	5.00	5.00	5.00	5.00	5.00	5.00	R1
Protocol 3	5.00	5.00	5.00	5.00	5.00	5.00	
Protocol 5	5.00	5.00	5.00	5.00	5.00	5.00	
Protocol 1	5.00	5.00	5.00	5.00	5.00	5.00	R2
Protocol 3	5.00	5.00	5.00	5.00	5.00	5.00	
Protocol 5	5.00	5.00	5.00	5.00	5.00	5.00	

**Table 10 diagnostics-15-01450-t010:** Kruskal–Wallis statistical comparison of DAP values across protocol settings for both chest and pelvis protocols.

Protocol Settings	Test Statistic	Std. Error	Std. Test Statistic	*p*-Values	Protocols
Protocol 3 vs. Protocol 2	8.00	3.878	2.06	**0.039**	Chest
Protocol 3 vs. Protocol 1	19.00	3.878	4.90	**<0.001**	
Protocol 2 vs. Protocol 1	11.00	3.878	2.84	**0.005**	
Protocol 3 vs. Protocol 5	−10.00	6.39	−1.57	0.117	Pelvis
Protocol 3 vs. Protocol 1	20.00	6.39	3.13	**0.002**	
Protocol 3 vs. Protocol 4	−29.50	6.56	−4.50	**<0.001**	
Protocol 3 vs. Protocol 2	39.00	6.39	6.11	**<0.001**	
Protocol 5 vs. Protocol 1	10.00	6.39	1.57	0.117	
Protocol 5 vs. Protocol 4	19.50	6.56	2.97	**0.003**	
Protocol 5 vs. Protocol 2	29.00	6.39	4.54	**<0.001**	
Protocol 1 vs. Protocol 4	−9.50	6.56	−1.45	0.148	
Protocol 1 vs. Protocol 2	−19.00	6.39	−2.97	**0.003**	
Protocol 4 vs. Protocol 2	9.50	6.56	1.45	0.148	

**Table 11 diagnostics-15-01450-t011:** Pairwise statistical comparison of quantitative image quality (*IQF_inv_*) across the three protocol settings for chest imaging.

Protocols	Test Statistic	Std. Error	Std. Test Statistic	*p*-Values
Protocol 3 vs. Protocol 2	12.45	3.93	3.16	**0.002**
Protocol 3 vs. Protocol 1	17.25	3.93	4.38	**<0.001**
Protocol 2 vs. Protocol 1	4.80	3.93	1.22	0.220

**Table 12 diagnostics-15-01450-t012:** Statistical pairwise comparison of the quantitative image quality metric *IQF_inv_* between the five pelvis protocol settings.

Protocol Settings	Test Statistic	Std. Error	Std. Test Statistic	*p*-Values
Protocol 1 vs. Protocol 3	−0.40	6.39	−0.06	0.950
Protocol 1 vs. Protocol 5	−19.20	6.39	−3.00	**0.003**
Protocol 1 vs. Protocol 2	−25.90	6.39	−4.05	**<0.001**
Protocol 1 vs. Protocol 4	−29.48	6.56	−4.49	**<0.001**
Protocol 3 vs. Protocol 5	−18.80	6.39	−2.94	**0.003**
Protocol 3 vs. Protocol 2	25.50	6.39	3.99	**<0.001**
Protocol 3 vs. Protocol 4	−29.08	6.56	−4.43	**<0.001**
Protocol 5 vs. Protocol 2	6.70	6.39	1.05	0.294
Protocol 5 vs. Protocol 4	10.28	6.56	1.57	0.117
Protocol 2 vs. Protocol 4	−3.58	6.56	−0.54	0.586

## Data Availability

Results and data of this study were not provided in any public places. The phantom images, calculations, and reports were archived in a local personal computer hard-drive.

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
