# Peer review of "Standardisation and Optimisation of Chest and Pelvis X-Ray Imaging Protocols Across Multiple Radiography Systems in a Radiology Department"

_diagnostics, 2025, doi:10.3390/diagnostics15121450_

Round 1
Reviewer 1 Report
Comments and Suggestions for Authors
Manuscript ID: diagnostics-3605098
Title: Standardisation and Optimisation of Chest and Pelvis X-Ray 2 Imaging Protocols Across Multiple Radiography Systems in a 3 Radiology Department
Reviewer comments:
-
Page 2, lines 77-80, please provide more elements to contextualize the paragraph, for example the geographical location of the radiology department.
-
Page 3, line 83, please explicitly state that the standardization of protocols is within a single center.
-
Page 4, line 104, in Table 1, please include the range of exposure time or tube charge (mAs) available for the X-ray equipment.
-
Page 5, line 119, please consider the use of variables and their units instead of just providing units. In this case, kV, mAs and mA require the corresponding variable. Please do not repeat the term "protocol" both in the heading and within the column. This also applies to Table 3 (page 7) and Table 7 (page 10).
-
Page 7, line 163, if possible, please improve the image quality of this figure, it seems to have a lower resolution compared to figure 2.
-
Page 7, line 175, please provide more details on the years of experience of the image evaluators, the term "several" may lead to ambiguity and make future comparisons of other studies with the present work difficult.
-
Page 7, line 180, please provide relevant technical data of the monitor used for visual image assessment.
-
Page 9, lines 216-219, It is relevant to report which test was used to assess the normality distribution and the p-value obtained.
-
Page 10, lines 231-148, The results section is somewhat confusing due to the interspersed presentation of thoracic and pelvic information. It is suggested that the results be separated by examination, with the analyses presented separately according to the area examined.
-
Pages 13 to 15, Tables 10 to 12, please indicate in the table those rows that show significant differences in their p-value; asterisks can be used to visually highlight this situation and facilitate the reader's analysis.
- Please include a discussion on inter-observer variability, such as calculating correlation coefficients, to validate your findings from the Visual Grading Analysis (VGA) method.
-
In the Discussion section, it is essential to include comparative data with other optimization studies to assess the magnitude of the dose reduction from the proposed strategy versus the same or similar strategies reported in the literature.
-
Please describe future lines of research or perspectives that emerge from your work, to build a line of work as a scientific community.
-
The conclusion could benefit from an approach more closely linked to the purpose of the study, since in its present form, it seems a brief summary, rather than a forceful closure of the work done.
-
Please check in the list of references those titles that appear in all capital letters, so that their presentation is consistent with the others.
Author Response
The response to the reviewer is provided in the attached separate Word document. The revised sections of the manuscript are marked in blue text

Reviewer 2 Report
Comments and Suggestions for Authors
The article I reviewed addresses the standardization and optimization of chest and pelvis X-ray protocols in the radiology department with a technically strong and clear presentation, with both quantitative and visual quality assessments. The study is of high value due to its methodological diversity and ALARA principle-compliant approach. However, some minor corrections are suggested before publication:
In the “Materials and Methods” section of the article, it is seen that all observers gave the highest score for the pelvis protocols in the VGA assessment. Since this situation may limit the discrimination of the assessment, it would be beneficial for methodological transparency to briefly mention the possible “ceiling effect” or reasons such as low observer variation in the discussion section.
The reason for excluding Protocols 2 and 4 from the VGA assessment is stated as “high dose”, but no comment is made on how high this dose is compared to acceptable threshold values. A brief explanation of the dose limit or the clinical context of the decision made in this section would increase the integrity of the study.
It is stated that the VGA assessment was performed with the ViewDEX platform, but in the visual scoring, the lighting conditions under which the assessment was made are only generally given as “compatible with the clinical environment”. When details such as the calibration status or the brightness level of the monitor on which the assessment was made are specified, the reliability of the results will increase.
Although the applicability and clinical benefit of the protocols are clearly stated in the conclusion section, a one-sentence guidance regarding the evaluation of the applicability of these protocols in a real patient population would be useful in the future. Such a suggestion emphasizes the clinical validity of the study.
In general, this study provides a solution to a clinically important need with scientific methods. It is suitable for publication if the minor revisions mentioned above are made.
Author Response

(The authors gave the same response as above.)

Reviewer 3 Report
Comments and Suggestions for Authors
Thank you for your immense effort that has been put in to this research, different methodologies to evaluate image quality were used, three different phantoms were used, subjective and objective evaluation tools, and observers of different proficiency.
1- Since all systems were found to operate on the same software platform and utilize the same image processing software moreover the systems share identical detector technology and X-ray tubes, were there any considerations to use different systems that differ in the technology? This would truly be the challenge!
2 -Table 3, SID for chest protocol 3? As known maximum to be used is 180 or 200 if you could justify the use of 300 in this protocol.
3 -Are there any significant differences between the scoring of radiographers and radiologists?
4- Regarding the used phantom for CTRAD, was any thickness used and what thicknesses were used to resemble soft tissue?
5- Moreover has the automated selection of tube current mAs been considered in this study ?
6- Justify the standardization of exposure parameters amog the same software platform that utilise the same image processing software moreover share identical detector technology and X-ray tubes.
Author Response
The response to the reviewer is provided in the attached separate Word document. The revised sections of the manuscript are marked in blue text.

Reviewer 4 Report
Comments and Suggestions for Authors
General comment: this study addresses the issue of variability in X-ray imaging protocols (chest and pelvis) across multiple radiography systems within a single radiology department. The primary aim was to harmonize and optimize these protocols by standardizing exposure parameters and geometric setups to minimize patient radiation dose while maintaining diagnostic image quality, adhering to the ALARA principle. The minor revisions suggested primarily aim to enhance clarity, provide slightly more context, and strengthen the discussion of specific findings (particularly the perfect pelvis VGA scores and comparison with external benchmarks).
Detailed comments:
- Introduction: consider briefly mentioning why such variability often exists in departments (e.g., historical reasons, vendor defaults, individual preferences) to strengthen the background.
- Methods:
- While mentioned it was from a previous thesis, briefly stating the reason for its development (e.g., "developed to explore dose reduction potential") might add context.
a. Specify the "average patient size" the PMMA plates/anthropomorphic phantoms are intended to simulate (e.g., in cm thickness or weight equivalent), if available.
b. State explicitly whether the observers performing the VGA were blinded to the protocol settings associated with each image. This is standard practice and important for mitigating bias.
c. Briefly justify the selection of the specific VGA criteria (Tables 5 & 6) – e.g., based on established guidelines (like the cited European Guidelines) or clinical importance.
- Results: adding a brief sentence summarizing the main finding after each table/figure within the Results section itself could improve readability
- Discussion:
- Please discuss the implications of the perfect VGA scores for the pelvis protocols more thoroughly. Did this make the selection of Protocol 3 solely based on dose and quantitative IQ, as visual quality was indistinguishable?
- Please discuss how the Automatic Exposure Control (AEC) system is expected to perform with the newly standardized protocols across the patient population.
- Please briefly compare the final selected DAP values or results with established diagnostic reference levels (DRLs) or other optimization studies, provide some additional references.
Author Response

(The authors gave the same response as above.)

Round 2
Reviewer 1 Report
Comments and Suggestions for Authors
Dear authors,
Thank you for addressing the observations from the previous review. I consider all requirements have been satisfactorily addressed, and this revised version presents the work with greater clarity and detail. In this context, I would like to propose a few specific observations to resolve minor details, primarily regarding formatting:
-
In Table 1, please correct the spelling of "load" (currently written as "loard") and add a hyphen to indicate the mAs range (0.5 – 800, instead of "0.5 800").
-
There appears to be a duplication issue in the paragraph: "The CDRAD phantom consists of a 15 × 15 array of cells with cylindrical holes of varying sizes (0.3 to 8.0 mm) and depths. The depth is constant within each column, and the area is constant within each row. Figure 1 provides a photographic representation andcan X-ray image of the CDRAD 2.0 phantom", it appears twice (lines 148-151, page 5 and lines 156-159, page 6).
-
Please remove the quotation marks at the end of the paragraph in line 180, page 7.
-
At the beginning of page 7 (lines 178-180) where the pelvis phantom is described, please include brand and model. Similarly, please consider adding brand and model information for the thorax phantom.
-
Please remove the capitalization of "Monthly" in line 216, page 8.
Author Response
Response to the reviewer and second revision of the manuscript are attached.
